# Depression and Anxiety-Free Life Expectancy by Sex and Urban–Rural Areas in Jiangxi, China in 2013 and 2018

**DOI:** 10.3390/ijerph18041991

**Published:** 2021-02-18

**Authors:** Yuhang Wu, Huilie Zheng, Zhitao Liu, Shengwei Wang, Xiaoyun Chen, Huiqiang Yu, Yong Liu, Songbo Hu

**Affiliations:** Jiangxi Province Key Laboratory of Preventive Medicine, School of Public Health, Nanchang University, Nanchang 330006, Jiangxi, China; 411437819016@email.ncu.edu.cn (Y.W.); zhenghuilie@ncu.edu.cn (H.Z.); 401440319007@email.ncu.edu.cn (Z.L.); 411437818006@email.ncu.edu.cn (S.W.); 411437820003@email.ncu.edu.cn (X.C.); yuhuiqiang@ncu.edu.cn (H.Y.); ly070310@ncu.edu.cn (Y.L.)

**Keywords:** healthy life expectancy, depression, anxiety, mental health, urban-rural areas

## Abstract

**Objective:** To quantitatively estimate life expectancy (LE) and depression and anxiety-free life expectancy (DAFLE) for the years 2013 and 2018 in Jiangxi Province, China, by sex and urban–rural areas. Additionally, to compare the discrepancy of DAFLE/LE of different sexes and urban-rural areas over various years. **Methods:** Based on the summary of the health statistics of Jiangxi Province in 2013 and 2018 and the results of the 5th and 6th National Health Service Surveys in Jiangxi Province, the model life table is used to estimate the age-specific mortality rate by sex and urban–rural areas. Sullivan’s method was used to calculate DAFLE. **Results:** Data from 2013 indicate that those aged 15 can expect to live 56.20 years without depression and anxiety for men and 59.67 years without depression and anxiety for women. Compared to 2013, DAFLE had not fluctuated significantly in 2018. The proportion of life expectancy without depression and anxiety (DAFLE/LE) declined between 2013 and 2018. DAFLE/LE in urban areas was higher than in rural areas. Men had higher DAFLE/LE than women. From 2013 to 2018, the DAFLE aged 15 decreased by 0.18 years for urban men and decreased by 0.52 years for urban women, rural areas also decreased to varying degrees. **Conclusions:** Even if women had a longer life span than men, they would spend more time with depression or anxiety. DAFLE did not increase with the increase in LE from 2013 to 2018, suggesting an absolute expansion of the burden, especially in rural areas. Depression and anxiety health services in Jiangxi, China will face more serious obstacles and challenges, which may lead to more disability. This requires more attention and more effective measures from the public, medical departments and the government.

## 1. Introduction

Mental disorders have become an increasingly important public health problem worldwide [1]. Depression and anxiety are the most common mental disorders worldwide [2,3,4,5]. Not only the diagnosis of anxiety and depression tends to occur at the same time, but their symptoms are also highly correlated [6]. It is estimated that in 2015, about 322 million people worldwide were suffering from depression, accounting for 4.4% of the global population, and about 260 million people suffering from anxiety disorders, accounting for 3.6% of the global population, many people suffer from both depression and anxiety [4]. It is not just the elderly [7,8,9] who are in their dying years, many adolescents [10,11] who have been in school for a long time and young people [12] who work every day are also troubled by depression and anxiety. The occurrence of these common mental disorders showed a rising trend, particularly acute in low-income countries [4].

Although life expectancy (LE) is getting longer, the burden of mental disorders is increasing [13]. In some developing countries, such as China, the burden of mental disorders even exceeds the burden of infectious diseases, heart disease, tumor diseases, and chronic diseases such as diabetes, hypertension, and obesity, becoming the most important disease cause for men and women [14,15]. Women are more likely to suffer from mental illness than men [3]. Mental disorders are one of the leading causes of disability and premature death worldwide [5]. Globally, depressive disorders are the third most frequent cause of adolescent disability-adjusted life-years lost, and anxiety disorders are the fifth most frequent cause of disability-adjusted life-years lost for adolescent girls [16]. The global community is speeding up the process of population aging. Depression will cause disability in the elderly, cognitive decline, functional impairment, and increase mortality and morbidity in the population [8,9,17]. The life expectancy gap between people with mental disorders and the general population persists and may even be widening [18]. A study reported that at age 18, quality-adjusted life expectancy (QALE) was 28.0 more years for depressed adults and 56.8 more years for non-depressed adults, a 28.9-year QALE loss due to depression in the US [19]. A Canadian study reported that men and women who had depression live a substantially higher proportion of their life in an unhealthy state compared to their counterparts without depression [20]. Another research showed that the negative correlation between anxiety and life expectancy required using appropriate methods to improve life expectancy [21]. Loss of life expectancy in most people with mental disorders such as depression or anxiety is accounted for by a broad range of causes of death [22,23,24].

Patients with depression and anxiety will not die in a short time, but are more likely to live with the disease. It is obviously not suitable to estimate the health status of the population solely based on the life expectancy at the end of death. Healthy life expectancy, based on physical function and health indicators, is a better indicator of people’s quality of life. In Brazil, Andrade [25] estimated life expectancy with and without depression among older adults for the years 2000 and 2010. The limitation of this study is that it involves elderly people aged 60 and above, and its results cannot yet reflect the reality of other lower age groups. In addition, the relationship between the mental health of residents and their place of residence(urban-rural areas) has been widely concerned by various countries [26]. However, the epidemiological evidence for this relationship has not been definitively concluded. An epidemiological study in England showed that urban environments increase the risk of anxiety, depression and psychosis [27]. A study from Denmark reported that people born in urban areas have a higher incidence of mental illness than people born in rural areas [28]. However, there are some opposite findings. No difference had been reported in the prevalence of mental disorders in most urban and rural areas of the United States [29]. Older Chinese living in rural areas reported more depressive symptoms than their urban counterparts [30].

In China, previous studies have focused on the analysis of the prevalence of depression or anxiety and related influencing factors. To the best of our knowledge, few studies have estimated healthy life expectancy associated with depression and anxiety. This information helps to more accurately locate the reality of depression and anxiety in China, a super populous country. The government and the public can better understand and reasonably respond to the burden of this rapidly aging society.

In this study, depression and anxiety-free life expectancy (DAFLE) and its radio to life expectancy (DAFLE/LE) of age groups were calculated in Jiangxi Province, China by sex and urban–rural areas for the years 2013 and 2018. We used DAFLE/LE as an indicator to evaluate the quality of life of the population in Jiangxi Province, China, which can reflect the impact of depression and anxiety on the quality of life to a certain exten. Following this, depression and anxiety life expectancy (DALE) and DAFLE were compared, respectively from 2013 to 2018.

## 2. Materials and Methods

### 2.1. Participants

This study analyzed data from the fifth and sixth National Health Service Survey in China in 2013 and 2018 by sex and urban–-rural areas. The National Health Service Survey in China was a nationwide cross-sectional survey to comprehensively understand residents’ health, health service needs and utilization, which was organized by the National Health Commission of China every fifth years since 1993. Details of the study sampling methodology can be found elsewhere [31]. In 2013, we obtained 8797 valid responses from 11,252 participants to answer the question (men vs. women: 4253 vs. 4544, urban vs. rural: 4623 vs. 4174). We learned from the original questionnaire that this question was only asked and answered for people aged 15 and above. Therefore, questionnaires completed for people under 15 years of age were excluded. The age group was divided by 10-year age groups with a last closed year at 75 years. In 2018, 7943 valid sample sizes were obtained from 10,123 participants (men vs. women: 3786 vs. 4157, urban vs. rural: 4147 vs. 3796).

### 2.2. Materials

Our study mainly requires the following two kinds of raw data:

In order to comprehensively evaluate the development of health services in Jiangxi Province, the Health Commission of Jiangxi Province collects and organizes relevant data every year and records them in the book “the Summary of Health Statistics of Jiangxi Province”. The annual data are separate documents for internal use only. Therefore, based on the Summary of Health Statistics of Jiangxi Province (2013 and 2018) [32,33], we obtained data on infant mortality rate (IMR) and under-five mortality rate (U5MR) by sex and urban-rural areas, and the data come from the maternal and child monitoring system. 

The prevalence data of depression and anxiety used in this study were from the fifth and sixth National Health Service Survey in China in 2013 and 2018. Donghu District, Zhanggong District, Yuanzhou District, Shanggao County, Gaoan County, and Panyang County were selected as sample counties (cities, districts) representing the overall urban and rural areas of Jiangxi Province. The research objects were classified into urban population and rural population based on their domicile place. The investigators comprised health personnel from county (city, district) health institutions and township health centres or community health service centres with specific professional knowledge. As face-to-face surveys, the investigators asked all participants of the survey household one by one according to the items in the questionnaires after being uniformly trained by the Statistics Information Center of the Health and Family Planning Commission. In this questionnaire, the health status of participants’ depression and anxiety is described for the question of whether they are anxious or depressed. Before the participants choose the answers based on their own understanding, the investigator will read the questions and provide the participants with appropriate explanations in accordance with the ICD-10 diagnostic criteria for depression and anxiety. There are three options for participants to choose from. If they answered “Do not feel anxious or depressed”, they are considered not to suffer from depression or anxiety. If they answered “consciously moderate anxiety or depression” or “consciously extreme anxiety or depression”, they are combined and considered to have anxiety or depression. It should be noted that no matter whether individuals feel depressed or anxious, or comorbidities, we treat them as suffering from depression and anxiety. 

### 2.3. Methods

On the basis of mortality data, we used the China model life table [34] method to estimate age-specific mortality rates and life expectancy by sex and urban-rural areas in Jiangxi Province for the years 2013 and 2018. The idea and principle of this China model table system are basically similar to the Murray model life table system [35], but which is constructed by using the census and population 1% Sample survey of China.

Sullivan’s method [36] is the most common method for calculating healthy life expectancy. The formula is as follows:$$HLE_{x}=\frac{1}{\ell_{x}}\overset{x_{\max}}{\underset{x}{\sum }}(L_{x}*\prod_{x})$$

In the above formula, $\ell_{x}$ and $L_{x}$ are the number of survivors and person-years in the corresponding age group in the life table. $\prod_{x}$ is a newly added indicator item in the life table, which represents the health status of depression and anxiety among people of different ages. 

The life table is used to estimate the life expectancy of each age group, which is further divided into unhealthy life expectancy and healthy life expectancy. In order to compare the differences by sex and urban-rural areas between 2013 and 2018, we calculated DAFLE/LE, which is the ratio of life expectancy without depression and anxiety to life expectancy. The standard error of the depression and anxiety rate was used to calculate the standard error of the DAFLE to calculate the corresponding 95% confidence intervals (CI) [37]. Linear regression analysis for each sex was used to test the age trend of DAFLE/LE. The Wilcoxon Signed Ranks test was used to explain the difference between man and women, urban and rural areas and 2013 and 2018 for DAFLE/LE.

## 3. Results

Table 1 shows the results related to LE, DAFLE and the proportion of DAFLE by sex and age in 2013 and 2018 in Jiangxi, China. In 2013, the LE for those aged 15 was 58.78 years for men and can expect to live 56.20 years without depression and anxiety. For women, these figures were 63.26 years and 59.67 years, respectively, both of which were higher than those for men. In 2018, the LE for those aged 15 increased to 60.35 years for men and 64.78 years for women. They can expect to live 56.26 years without depression and anxiety for men and 59.66 years for women.

Between 2013 and 2018, the proportion of DAFLE at the age of 15 for men decreased by about 2.4% from 95.6% to 93.2%. Even at the age of 75 and above, this proportion still dropped by 2.7% (88.9% to 86.2%). Among Jiangxi women, for those aged 15 years, there was a 2.2% decrease (from 94.3% to 92.1%) in the proportion of DAFLE. However, the proportion of DAFLE for women aged 75 and above in their remaining life increased from 85.0% to 85.3%. The results of linear regression analysis and the Wilcoxon Signed Ranks test showed that all of the *p* values were less than 0.05. 

Figure 1 presents the absolute difference in DALE and DAFLE between 2013 and 2018. the DALE at the age of 15 years for men increased by 1.50 years and the DAFLE only increased by 0.07 years. At the age of 75 and above, the increase in DALE and DAFLE was 0.31 years and 0.24 years, respectively. Among Jiangxi women, for those aged 15 years, there was a 1.53-year increase in DALE, and a 0.01-year decrease in DAFLE. However, DALE increased by 0.06 years in women aged 75 and above, and DAFLE increased by 0.55 years.

Table 2 divides the total sample into urban and rural residents and describes their results. People in urban areas had longer LE and DAFLE than those in rural areas. In 2013, urban men at age 15 were expected to live 58.82 years and 55.30 years for rural men. For women at age 15, DAFLE was 63.09 years in urban areas and 57.88 years in rural areas. DAFLE/LE was 96.5% for men and 96.4% for women aged 15 in urban areas, which was higher than the same age group in rural areas (men: 94.4%, women: 91.9%). The same is true for 2018.

Figure 2 presents the absolute changes (2018–2013) in DALE and DAFLE by sex and urban-rural areas between 2013 and 2018. From 2013 to 2018, the DAFLE at the age of 15 decreased by 0.18 years for urban men and decreased by 0.52 years for urban women. Similar to the results in urban areas, the life expectancy of rural men and women without depression and anxiety also decreased to varying degrees. Among for men and women aged 15, there was a 1.05-year increase in urban men, 2.04-year increase in rural men and 1.35-year increase in urban women, 1.78-year increase in rural women in DALE, suggesting an absolute expansion of the number of years lived with depression and anxiety in Jiangxi, China, especially in rural areas.

## 4. Discussion

Women have a longer life expectancy and healthy life expectancy than men. However, women were more likely to spend a higher proportion of their remaining lives with depression and anxiety than men. Generally speaking, the length of life displayed by women is more advantageous than that of men. In our study, women have a higher prevalence of depression and anxiety than men (2013: 5.3% vs. 4.3%, 2018: 8.3% vs. 7.2%). The rates of depression and anxiety in women are up to double those in men [38]. Under the dual effect of morbidity and mortality, women will live longer with disabilities, other mental health problems, and some chronic diseases caused by depression and anxiety. They cannot maintain a higher level of quality of life, even worse than men who do not have the advantage of length of life. 

From 2013 to 2018, With the increase in LE, DAFLE did not increase significantly. Jiangxi experienced both an absolute and a relative expansion in depression and anxiety: DALE increased significantly and the proportion of life expectancy without depression and anxiety declined for most age groups between 2013 and 2018. During these five years, China’s economy and social health services have developed rapidly, reflecting the increase in life expectancy of the population. However, the mental health of the people in Jiangxi Province is getting worse and worse. The main source of depression and anxiety among Chinese adolescents is related to school pressure [39]. Influenced by the Chinese education system, the success of young people is largely measured by examinations [40], exams are especially frequent in schools. In addition, in China, teenagers are more vulnerable to school violence, abuse and bullying, which seriously affects their mental health [41,42]. Young and middle-aged people tend to focus on work and dealing with their daily needs for food, clothing, housing and transportation. Such a high-intensity life rhythm and intense distraction may cause mental disorders and delay the manifestation of symptoms. Furthermore, Network expression is an urgent issue in China. DAFLE/LE decreased over time and age, which suggested that with older people living longer, they will spend a greater proportion of their life with depression or anxiety. Elderly people begin to experience a physiological decline in various aspects of body functions, which is more likely to be accompanied by depression and anxiety-related diseases. Elderly people’s self-esteem and social status decline, which leads to a negative view of the aging process and induces depressive symptoms [43]. In China, people prefer to seek treatment from the medical department when they feel discomfort or pain in the body, while the mental health status is manifested as a low perceived need. In other words, most Chinese people do not think they should be treated [44], even if they are diagnosed with depression or anxiety. This may be caused by patients and family members’ lack of knowledge about mental disorders or fear of stigma associated with seeking care from a psychiatric service [45].

Affectivity should be received sufficient attention, especially negative affect [46]. Variables including biological, temperamental, psycho-social, socio-contextual, and environmental ones can affect human affectivity [47,48]. It is also important to enhance individual self-esteem and social support in daily life [46]. Connecting depression and anxiety with affectivity, self-esteem and social support cannot only help depressed or anxious patients to better deal with their lives, but also may indicate the onset of depression and anxiety, which will be of great significance to the prevention of mental disorders.

Promoting mental health is the focus of public health in the future. Future measures should emphasize the comprehensive promotion of mental health care for all age groups from the three perspectives of early detection, early diagnosis, and early treatment with the necessary support of society and policy makers. School-based academic mitigation programs and anti-school violence programs are effective in reducing subsequent aggression or internalization problems among adolescents [49]. An economic argument can be made to employers to motivate them to intervene proactively to reduce stress and improve working environments [50]. It is very necessary to build a complete social support network for the elderly. It provides individuals with opportunities to establish friendship, attachment and intimacy with others, helping them to release negative emotions during the aging process [9]. Mental health treatment urgently needs to be included in further comprehensive patient medical management. In China, the integration of mental health care with other chronic disease care and primary health care services is a strategically significant move [45]. From 2013 to 2018, the increased in unhealthy life expectancy suggested that clinical management and treatment of patients with depression and anxiety should be strengthened. The patients who had other diseases associate with depression and anxiety were given a simple psychological test or filled out a scale to measure depression and anxiety, so as to initially screen out high-risk groups with depression and anxiety, which will help improve the efficiency of the medical service system. Stigma underlies many of the barriers to healthcare access [51]. Awareness campaigns about mental health issues are essential to encourage those who are suffering silently to seek medical care and speak out about their diseases [3].

We conducted a discussion based on urban and rural areas to rationally guide the allocation of limited health resources. The results of sex differences in urban and rural areas are consistent with those described above. In Jiangxi Province, people living in urban areas (whether men or women) had higher LE, DAFLE and can expect to live a higher proportion of their remaining lives with depression and anxiety than rural areas. From 2013 to 2018, at younger ages (i.e., 15 and 25 years), Increased life expectancy is offset by the same increasing years of depression and anxiety, and even life expectancy without depression and anxiety has decreased. Obviously, in China, urban residents enjoy more conveniences, such as more health resources, a more complete education model and a more comfortable living environment [52]. Chinese rural residents may have to face problems such as lagging social demographic development, low level of education, few employment opportunities, poor health and social infrastructure, and economic difficulties [53]. Another study found that the utilization rate of mental health services such as hospitalization expenses, length of stay and frequency of hospitalization among rural residents in China is lower than that of urban residents [54]. 

China’s National Mental Health Law, effective from May, 2013 [45], shifting the focus of services from specialized psychiatric hospitals in urban centers to general hospitals, communities, and urban and rural social health centers, the treatment and support for patients with mental disorders is ensured. This is more to expand the elderly’s access to mental health services. In fact, under the influence of this policy, the depression and anxiety of the elderly have indeed improved. For the high-age groups (65 years, 75 years and above), life expectancy without depression and anxiety has increased to varying degrees between 2013 and 2018(except for rural men aged 75 and above). However, at the age of 75, the proportion of life expectancy without depression and anxiety in rural areas has dropped significantly than in urban areas. Compared with urban elderly, the government’s support in protecting rural elderly from depression and anxiety is negligible, older adults in rural areas receive fewer pension and welfare delivery from the government, saying nothing of mental health services [9]. The tradition of respecting, valuing and caring for the elderly is deeply rooted in China. However, with the development of urbanization and the implementation of the one-child policy in the past, a large number of laborers from rural areas have flowed into urban areas, which has further promoted the separation of the young generation from their elderly parents [9], resulting in a large number of empty-nest elderly in rural areas. Social separation and social isolation are closely related to depression and anxiety [55].

The government should also focus on reducing the gap of mental health coverage and resource allocation between urban and rural areas. In addition to being given adequate social support, rural residents should also be given more medical resources and social infrastructure. The successful management and treatment of depression and anxiety disorders in terms of infrastructure, resources, personnel and training programs require the joint efforts of doctors, nurses, psychologists, social workers and volunteers [8]. Experts from urban areas are able to supervise non-professional workers and provide advice to depression and anxiety patients in rural areas through a mobile treatment team [56].

There are some limitations in this study. First, Questionnaires about depression and anxiety are self-reports, not clinical diagnoses. Data based on self-reports implies the possibility of deviation. If anxiety and depression are clinically evaluated and diagnosed, the results may be different. However, these results emphasize the impact of depression and anxiety disorders on the remaining quality of life. Second, the healthy life expectancy calculated by Sullivan’s method only uses the cross-sectional prevalence. The mechanisms related to depression and anxiety were not explored in this study. Although urban and rural areas contain related variables, these variables change over time. The interaction between anxiety and depression can be complex. Therefore, it is difficult to measure with a single scale. Future research may consider using an effective scale in combination, which may provide a clearer picture.

Third, unfortunately, COVID-19 has changed everything, and it also has a serious impact on the depression and anxiety of the population. Therefore, the status and trends of depression and anxiety in 2013 and 2018 cannot explain the current situation. We should compare the data of this special period with the previous data in 2013 and 2018 in future research to reveal the impact of COVID-19 on DAFLE.

## 5. Conclusions

Even though women live longer lives, they live more years with depression and anxiety disorders. From 2013 to 2018, life expectancy with depression and anxiety increased significantly and the proportion of life expectancy without depression and anxiety declined in Jiangxi, which may reflect living more years with lower quality of life and worse health status. The elderly in rural areas are particularly affected by depression and anxiety. Mental health problems need to be paid attention to and widely supported by the public, medical departments the government. At the same time, the findings have certain reference significance for the development of mental health work in other countries and regions similar to Jiangxi Province, China.

## Figures and Tables

**Figure 1 ijerph-18-01991-f001:**
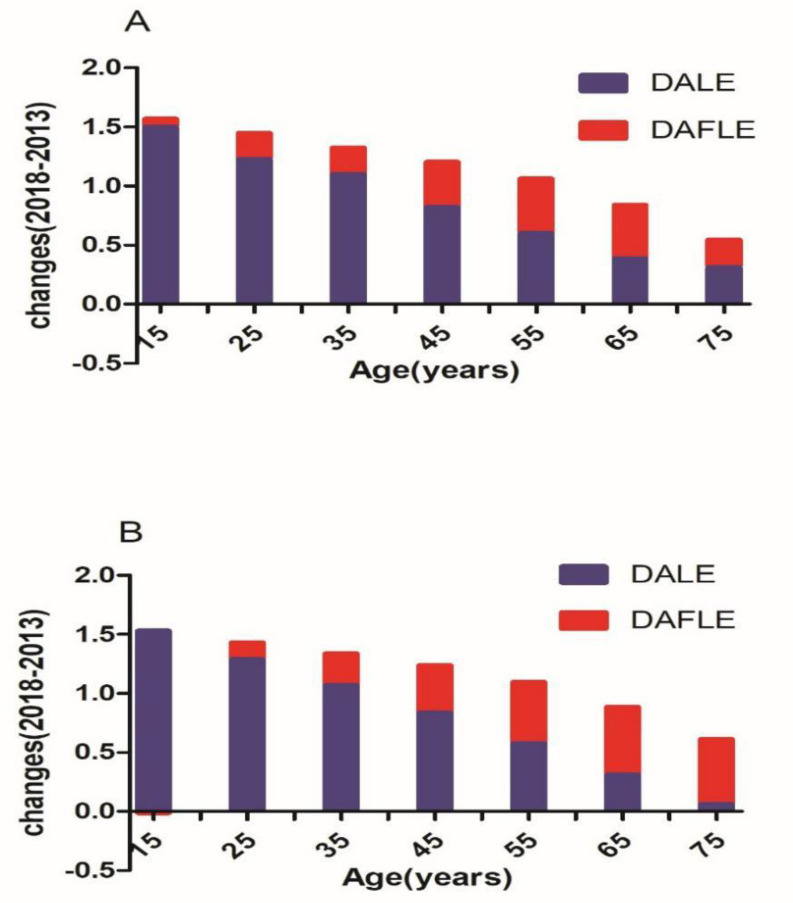
Absolute changes in depression and anxiety life expectancy (DALE) and depression and anxiety-free life expectancy (DAFLE) by sex at various ages in Jiangxi, China between 2013 and 2018 (**A**). men (**B**). women.

**Figure 2 ijerph-18-01991-f002:**
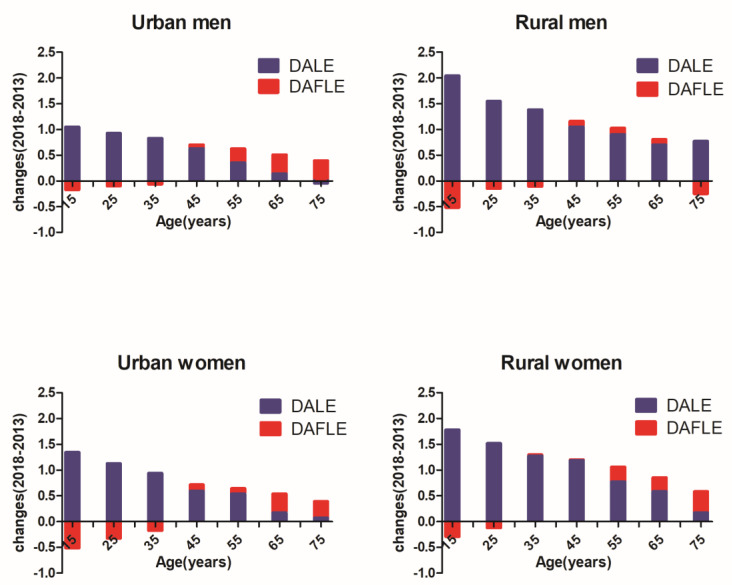
Absolute changes in depression and anxiety life expectancy (DALE) and depression and anxiety-free life expectancy (DAFLE) by sex and urban-rural areas in Jiangxi, China for the years 2013 and 2018.

**Table 1 ijerph-18-01991-t001:** LE, DAFLE and the proportion of DAFLE in Jiangxi, China, by sex and age for the years 2013 and 2018.

Age	Men			Women		
	LE	DAFLE (95%CI)	HALE/LE (%)	LE	DAFLE (95%CI)	DAFLE/LE (%)
2013						
15	58.78	56.20 (55.85, 56.54)	95.6	63.26	59.67 (59.24, 60.10)	94.3
25	49.29	46.74 (46.41, 47.08)	94.8	53.55	50.00 (49.57, 50.43)	93.4
35	39.89	37.52 (37.20, 37.83)	94.0	43.90	40.53 (40.12, 40.95)	92.3
45	30.74	28.59 (28.29, 28.89)	93.0	34.41	31.37 (30.97, 31.77)	91.2
55	22.15	20.28 (19.99, 20.58)	91.6	25.33	22.71 (22.32, 23.09)	89.6
65	14.53	13.05 (12.76, 13.35)	89.8	17.01	14.93 (14.55, 15.31)	87.8
75	8.66	7.70 (7.39, 8.02)	88.9	10.19	8.66 (8.26, 9.06)	85.0
2018						
15	60.35	56.26 (55.79, 56.74)	93.2	64.78	59.66 (59.13, 60.20)	92.1
25	50.74	46.96 (46.53, 47.40)	92.6	54.98	50.14 (49.65, 50.64)	91.2
35	41.21	37.74 (37.34, 38.13)	91.6	45.24	40.81 (40.35, 41.26)	90.2
45	31.95	28.97 (28.62, 29.32)	90.7	35.65	31.77 (31.35, 32.20)	89.1
55	23.21	20.74 (20.41, 21.07)	89.4	26.43	23.23 (22.83, 23.63)	87.9
65	15.37	13.51 (13.20, 13.82)	87.9	17.89	15.51 (15.13, 15.89)	86.7
75	9.21	7.94 (7.61, 8.26)	86.2	10.80	9.22 (8.83, 9.60)	85.3

LE, Life expectancy; DAFLE, Depression and anxiety-free life expectancy; 95% CI, 95% confidence intervals.

**Table 2 ijerph-18-01991-t002:** LE, DAFLE and the proportion of DAFLE for various ages in urban-rural areas.

Age	2013			2018		
	LE	DAFLE (95%CI)	DAFLE/LE (%)	LE	DAFLE (95%CI)	DAFLE/LE (%)
	Urban Men					
15	60.93	58.82 (58.38, 59.26)	96.5	61.80	58.64 (58.07, 59.21)	94.9
25	51.27	49.19 (48.76, 49.63)	95.9	52.10	49.09 (48.55, 49.63)	94.2
35	41.71	39.78 (39.37, 40.20)	95.4	42.48	39.72 (39.22, 40.22)	93.5
45	32.40	30.63 (30.22, 31.03)	94.5	33.10	30.70 (30.25, 31.15)	92.7
55	23.61	21.99 (21.59, 22.38)	93.1	24.24	22.26 (21.84, 22.68)	91.8
65	15.69	14.39 (14.00, 14.78)	91.7	16.20	14.75 (14.38, 15.13)	91.1
75	9.43	8.54 (8.14, 8.94)	90.6	9.78	8.94 (8.57, 9.30)	91.4
	Urban Women					
15	65.47	63.09 (62.58, 63.59)	96.4	66.30	62.57 (61.93, 63.20)	94.4
25	55.64	53.25 (52.75, 53.76)	95.7	56.44	52.92 (52.34, 53.51)	93.8
35	45.86	43.58 (43.08, 44.07)	95.0	46.62	43.40 (42.84, 43.95)	93.1
45	36.23	34.09 (33.60, 34.58)	94.1	36.95	34.21 (33.71, 34.71)	92.6
55	26.95	25.13 (24.66, 25.60)	93.3	27.59	25.23 (24.75, 25.71)	91.5
65	18.32	16.78 (16.31, 17.25)	91.6	18.86	17.15 (16.71, 17.58)	90.9
75	11.11	10.03 (9.56, 10.50)	90.3	11.50	10.35 (9.92, 10.79)	90.0
	Rural Men					
15	58.56	55.30 (54.73, 55.86)	94.4	60.08	54.78 (53.96, 55.59)	91.2
25	49.09	45.89 (45.33, 46.45)	93.5	50.49	45.74 (45.04, 46.44)	90.6
35	39.71	36.71 (36.18, 37.23)	92.4	40.98	36.60 (35.96, 37.24)	89.3
45	30.57	27.90 (27.41, 28.40)	91.3	31.74	28.02 (27.45, 28.58)	88.3
55	22.00	19.76 (19.28, 20.24)	89.8	23.03	19.88 (19.34, 20.43)	86.4
65	14.41	12.62 (12.13, 13.11)	87.6	15.22	12.73 (12.20, 13.26)	83.6
75	8.59	7.40 (6.84, 7.96)	86.1	9.11	7.14 (6.54, 7.75)	78.4
	Rural Women					
15	63.02	57.88 (57.13, 58.63)	91.9	64.50	57.59 (56.67, 58.51)	89.3
25	53.32	48.28 (47.54, 49.02)	90.5	54.72	48.15 (47.30, 49.00)	88.0
35	43.69	38.97 (38.26, 39.67)	89.2	44.99	38.99 (38.20, 39.78)	86.7
45	34.22	30.03 (29.35, 30.71)	87.8	35.42	30.05 (29.30, 30.79)	84.8
55	25.16	21.48 (20.81, 22.15)	85.4	26.22	21.77 (21.05, 22.49)	83.0
65	16.87	13.99 (13.32, 14.65)	82.9	17.72	14.26 (13.57, 14.96)	80.5
75	10.10	7.86 (7.15, 8.57)	77.8	10.68	8.27 (7.55, 9.00)	77.4

LE, Life expectancy; DAFLE, Depression and anxiety-free life expectancy; 95% CI, 95% confidence intervals.

## Data Availability

The data presented in this study are available on request from the corresponding author. The data are not publicly available due to privacy.

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
