# Peer review of "Depression and Anxiety-Free Life Expectancy by Sex and Urban–Rural Areas in Jiangxi, China in 2013 and 2018"

_ijerph, 2021, doi:10.3390/ijerph18041991_

Round 1

Reviewer 1 Report

Article

Depression and anxiety-free life expectancy by sex and urban-2 rural areas in Jiangxi, China in 2013 and 2018

The study estimates and compares life expectancy with and without depression and anxiety for the years 2013 and 2018 in Jiangxi province, China, by sex and urban–rural areas.

In general, I recommend moderate revisions as the text definitely needs a thorough correction by a native English speaker.

I recommend also to review the spaces between words, especially before the square brackets of citations (there’s no space before bracket).

The “Objective” in Abstract section isn’t clear, the Authors should correct and modify.

The analysis of anxious and depressive symptoms is not clear, the Authors should give more informations about questionnaires on depression and anxiety.

The data on rural and urban areas is interesting, the authors should describe what is meant by rural area and urban area (for example for rural area having a garden or living in the countryside).

The concept and the idea is interesting but the authors should add to limits that, unfortunately, COVID-19 has changed everything, even the aspects related to depressive and anxious symptomatology, therefore, talking about 2013 and 2018 can provide data not entirely truthful.

I encourage the authors to review in the future the data and compare them to current ones, possibly making a comparison between pre and post COVID-19, this topic can be write in “Future research” (line 299).

Author Response

Dear reviewer,

We wish to thank you for your precise and useful review. We are accepting all of suggested changes.

Response to Reviewer 1 Comments

Point 1: In general, I recommend moderate revisions as the text definitely needs a thorough correction by a native English speaker. I recommend also to review the spaces between words, especially before the square brackets of citations (there’s no space before bracket).

Thank you very much for your careful and useful review. We have reviewed the full text and revised it according to your suggestions. In addition, we had a discussion with the editor about when to polish the manuscript, and the editor believes that the manuscript will be revised by a native English speaker after making further decisions.

Point 2: The “Objective” in Abstract section isn’t clear, the Authors should correct and modify.

We totally agree with your review After we reviewed the manuscript, the results and discussion section analyses the discrepancy of DAFLE/LE of different sexes and urban-rural areas over various years (2013-2018), which should be described in the objective part of the abstract. Based on your comments, we made appropriate revisions: line 13-16

Point 3: The analysis of anxious and depressive symptoms is not clear, the Authors should give more information about questionnaires on depression and anxiety.

Thank you for providing us with such a useful comment. The diagnosis of anxiety and depression was more than just a question. We provided more information about how the research participants diagnosed with depression and anxiety in the manuscript (line 161-172). We also mentioned that for the sampling method (reference 31) and some details of the surveys (line 125-137).

Point 4: The data on rural and urban areas is interesting, the authors should describe what is meant by rural area and urban area (for example for rural area having a garden or living in the countryside).

Thanks. We agree with your suggestion that the manuscript should include more descriptions of urban and rural areas. According to the fifth and sixth National Health Service Survey, there were six counties (cities, districts) in Jiangxi Province selected as representative samples (three urban areas and three rural areas). In addition, as face-to-face surveys, the research participants were classified into urban population and rural population based on their domicile place (line 159-160).

Point 5: The concept and the idea is interesting but the authors should add to limits that, unfortunately, COVID-19 has changed everything, even the aspects related to depressive and anxious symptomatology, therefore, talking about 2013 and 2018 can provide data not entirely truthful.

I encourage the authors to review in the future the data and compare them to current ones, possibly making a comparison between pre and post COVID-19, this topic can be write in “Future research” (line 299).

Thank you for giving us such a constructive suggestion. We agree with you. Under the influence of COVID-19, the status and trends of depression and anxiety in 2013 and 2018 cannot explain the current situation. We proposed to compare the data of this special period with the previous data in 2013 and 2018 in future research to reveal the impact of COVID-19 on depression and anxiety-free life expectancy in the manuscript (line 383-387).

Sincerely,

Yuhang Wu

Reviewer 2 Report

I would like to thank you for the opportunity to review this manuscript. Theme is of great interest. However, this study has a number of important shortcomings that are detailed below:
Abstract: Conclusions are not consistent and do not really correspond with the results. How do you refer to quality of life and health status? Have these been measured?
Introduction lacks an argument and justification for the study. What do the authors want to show by relating both mental health problems to other health problems? These are not analyzed later. Are there previous studies?
Main deficiency of this study is the diagnosis of depression and anxiety. Has the population been diagnosed with a single question? How was the procedure for conducting the surveys carried out?
Statistical tests performed appear in the discussion, I advise that these tests are not included here as justification. On the other hand, an adequate discussion is not offered according to the results obtained.
How relevant is this study to clinical practice? It is not clear.

Author Response

Dear reviewer,

We wish to thank you for your precise and useful review. We are accepting all of suggested changes.

Response to Reviewer 2 Comments

Point 1: Abstract: Conclusions are not consistent and do not really correspond with the results. How do you refer to quality of life and health status? Have these been measured?

Thank you very much for your valuable suggestions. We agree with your review after we reviewed the manuscript. The results and conclusions of the abstract section have been revised (line 27-37).

We may not express this part clearly in our research. Therefore, we have made some revisions in our manuscript (line 118-121). In fact, we used the ratio of life expectancy without depression and anxiety to life expectancy (DAFLE/LE) as an indicator to evaluate the quality of life and health status of the population, which can reflect the impact of dementia on the quality of life and health status to a certain extent. The smaller the DAFLE/LE is, the worse the quality of life and health status due to depression and anxiety. This gap in healthy life expectancy between population of Jiangxi with and without depression and anxiety is primarily associated with losses in quality of life.

Point 2: Introduction lacks an argument and justification for the study. What do the authors want to show by relating both mental health problems to other health problems? These are not analyzed later. Are there previous studies?

Thank you very much for your careful and useful review. We agree with your comments. We have cited several arguments to describe that depression and anxiety are associated with life expectancy (line 74-85).

The reason why we try to link mental health problems with other health problems is that we believe that most people with mental disorders do not die from mental diseases such as depression or anxiety, so the decrease of their life expectancy is caused by other health problems associate with mental health problems. However, we found that the over-description of this connection seemed to deviate from the theme after we reviewed the manuscript. Therefore, we had appropriately deleted the content of the introduction part (line 85-92).

Point 3: Main deficiency of this study is the diagnosis of depression and anxiety. Has the population been diagnosed with a single question? How was the procedure for conducting the surveys carried out?

Thank you for providing us with such a useful comment. The diagnosis of anxiety and depression was more than just a question. We provided more information about how the research participants diagnosed with depression and anxiety in the manuscript (line 161-172). We also mentioned that for the sampling method (reference 31) and some details of the surveys (line 125-137).

Point 4: Statistical tests performed appear in the discussion, I advise that these tests are not included here as justification. On the other hand, an adequate discussion is not offered according to the results obtained.

Thanks. Yes, we agree with your suggestion: we revised the manuscript and transferred the statistical tests in the discussion section to the results (line 218-219).

According to the results obtained, we have revised and appropriately supplemented the discussion (line 265, 284-286). Paragraphs 1, 2, and 5 of the discussion parts described the differences of men-women, 2013-2018 and urban-rural areas, respectively.

The reason why some information did not seem to be provided for sufficient discussion was that after adding the dimension of urban-rural areas, some results may consistent with what had been described and discussed in the manuscript, such as sex differences. Therefore, we may not describe it again but. simply describe it in one sentence (line 329-330).

Point 5: How relevant is this study to clinical practice? It is not clear.

Thank you for your detailed comment.

i:) Regarding possible practical or clinical implications, our results suggested that it is necessary to consider the assessment of DAFLE from 2013 to 2018 as primary prevention and secondary prevention strategies (line 305-318).

ii:) Certain groups of people (such as rural areas) may have more severe psychological disorders, which can make the government implement targeted intervention measures (line 328-343).

iii:) From 2013 to 2018, the increase in unhealthy life expectancy suggested that clinical management and treatment of patients with depression and anxiety should be strengthened. (We added this point in the discussion section: line 318-320)

iv:) In addition, in the manuscript, we also proposed to carry out a simple psychological test or fill in a scale to measure the depression and anxiety of patients in the hospital, and preliminarily screen out the high-risk groups of depression and anxiety. In the future, it may alleviate the pressure brought by the growing burden of psychological disorders in the clinic to a certain extent (line 321-324).

Sincerely,

Yuhang Wu

Reviewer 3 Report

Variable - symptoms of anxiety and depression were obtained only through declarations. I believe that accurate and reliable research methods should have been used. This, in my opinion, greatly reduces the value of research. Only calculations based on percentages were performed. I believe that the significance of statistical differences should be counted.

Author Response

Dear reviewer,

We wish to thank you for your precise and useful review. We are accepting all of suggested changes.

Response to Reviewer 3 Comments

Point 1: Variable - symptoms of anxiety and depression were obtained only through declarations. I believe that accurate and reliable research methods should have been used. This, in my opinion, greatly reduces the value of research. Only calculations based on percentages were performed. I believe that the significance of statistical differences should be counted.

Thank you for providing us with such a useful comment. The diagnosis of anxiety and depression was more than just a question. We provided more information about how the research participants diagnosed with depression and anxiety in the manuscript (line 161-172). We also mentioned that for the sampling method (reference 37) and some details of the surveys (line 125-137).

i:) In the manuscript, we compared the gender difference and urban-rural difference through the estimation of DAFLE, and we calculated the 95% confidence interval of DAFLE. We did linear regression analysis to test whether the DAFLE/LE decreased with age. the results show that all of P values were less than 0.05, which indicated that with older age and over time, DAFLE/LE gradually decreased (line 200-204). 

ii:) DAFLE/LE as a relative indicator reflects the percentage of healthy life years (without depression or anxiety) in total life expectancy. In order to explain the difference of man-women, urban-rural areas and 2013-2018 for DAFLE/LE, statistical method of the Wilcoxon Signed Ranks test was used. We added the description of this part to the manuscript (line 200-204).

Sincerely,

Yuhang Wu

Reviewer 4 Report

Manuscript deals with Life Expectancy (LE) and its association with both self-reported anxiety and depression state. For that, institutional archives data were collected, in two different dates/years. Data are segregated by sex/gender and rural/urban context. I think this manuscript fits with the aims and scope of IJERPH.

As critical aspects, I can observe the following points:

  • First of all, a formal aspect: ‘Materials and Methods’ section has an unusual structure, and basically represents ‘procedure’ subsection. Despite data are extracted from institutional archives, I think a structure with ‘participants’, ‘materials’ etc… would be more appropriate for a scientific report. In this sense, I miss a better description of samples (N, sex, age ranks…), and the survey used.
  • I also miss a better justification why is relevant LE relates with anxiety/depression. As authors point out, these mental disorders have high prevalence rates and are frequently associated with relevant health problems (cancer, coronary…). But they are less disabling than others mental disorders (bipolar, schizophrenia, dementia …). LE could be different in these more disabling disorders.
  • Major concern is related with how anxiety / depression was measure: depression and anxiety are estimated by only one question, as part of a general survey. No more information is provided. We do not know what respondents understand when they are ‘feeling anxious or depressed’, and I cannot know if respondents refer to a situational current state or they are expressing how they generally feel… With this conditions, it is difficult to identify responses as a precise anxiety / depression measure. Both depression and anxiety states are (especially) more complex than a single item (about if respondents feel anxious or depression). There are several well-stablished measures for anxiety and depression, with adequate psychometric values. In ‘Discussion’ section authors recognize some limitations in this sense. But the argument about the use of self-reports is clearly insufficient. I think they can try to justify the opportunity of the use of one-question measures, and in which sense, those measures are precise (reliable and valid) to assess a target variable. This is relevant because of any argument about LE with or without depression / anxiety can be understand as arbitrary.
  • Most relevant data are presented in percentages. Those data are discussed about increments/decrements between the two years when data were collected, comparing sex / genders, rural / urban… I miss some statistical coefficient (OR, chi squared…) to stablish when, comparing percentages, they are really different (statistical significance).
  • When data are discussed, one of the points is about women have more LE and they are more anxious / depressed. I could not found a specific data about specific prevalence rate of anxiety and depression. It is well-known women attain higher rates than men in both disorders and I suspect this data is coincident with their surveys. Manuscript statement about how women have higher LE and higher anxiety / depression levels and, in this sense, women will live more time with anxiety / depression sound tautological. I am not sure there is a direct correspondence (correlation) between LE and the presence of emotional negative feeling.
  • Finally, result implications are discussed about the need of suitable health services and resources. I agree with authors about this necessity. But I think data can also be discussed in how people deal with their lives and how emotional negative feelings are present across life span and the need for a new educational paradigm that includes psychological resources related with the use of efficient emotion regulation strategies. This is especially relevant if it is true that those negative feelings are predicting the initial onset of anxiety and depression (e.g., doi:10.3390/ijerph17196984)

Author Response

Dear reviewer,

We wish to thank you for your precise and useful review. We are accepting all of suggested changes.

Response to Reviewer 4 Comments

Point 1: First of all, a formal aspect: ‘Materials and Methods’ section has an unusual structure, and basically represents ‘procedure’ subsection. Despite data are extracted from institutional archives, I think a structure with ‘participants’, ‘materials’ etc… would be more appropriate for a scientific report. In this sense, I miss a better description of samples (N, sex, age ranks…), and the survey used.

Thank you for providing us with such a useful comment. We followed your suggestions and made appropriate revisions to the materials and methods section (line 124-204).

i:) In general, we revised structure of ‘Materials and Methods’ into the structure you mentioned, which was more suitable for scientific reports.

ii:) We revised and adjusted the sample description.

Point 2: I also miss a better justification why is relevant LE relates with anxiety/depression. As authors point out, these mental disorders have high prevalence rates and are frequently associated with relevant health problems (cancer, coronary…). But they are less disabling than others mental disorders (bipolar, schizophrenia, dementia …). LE could be different in these more disabling disorders.

Thank you very much for your careful and useful review. We agree with your comments. We have cited several arguments to describe that depression and anxiety were associated with life expectancy (line 74-85).

I’m sorry we may not fully understand what you mean according to some comments of disabling. But I will try to explain to you.

i:) In general, there are two types of healthy life expectancy: healthy status life expectancy and healthy adjusted life expectancy. Our research is about healthy status life expectancy. In our study, the healthy life years of individuals considered to be suffering from depression or anxiety will be directly deducted. The data on life expectancy without depression and anxiety (DAFLE) was estimated by prevalence of depression and anxiety rather than disability rates. The greater the prevalence of depression and anxiety, the shorter the DAFLE. In fact, there are currently many studies on healthy status life expectancy.  (doi:10.1159/000381848, doi:10.1590/S1518-8787.201605000590)

ii:)What you mentioned may be a study on healthy adjusted life expectancy. Different diseases have different degrees of disability. Healthy adjustment life expectancy is related to the disability weight of diseases. If disability is included, we need to adjust the life table according to the disability weight of each age group. We regret that all calculations and analyses only consider mortality and prevalence, but not disability. At the same time, we are very grateful to you for providing ideas for our further research. The GBD study (doi:10.1016/S0140-6736(18)32335-3) was take into account the disability weight when reporting the healthy life expectancy in various regions of the world. We hope to refer to their method and take the disability into account in further studies.

Point 3: Major concern is related with how anxiety / depression was measure: depression and anxiety are estimated by only one question, as part of a general survey. No more information is provided. We do not know what respondents understand when they are ‘feeling anxious or depressed’, and I cannot know if respondents refer to a situational current state or they are expressing how they generally feel… With this conditions, it is difficult to identify responses as a precise anxiety / depression measure. Both depression and anxiety states are (especially) more complex than a single item (about if respondents feel anxious or depression). There are several well-stablished measures for anxiety and depression, with adequate psychometric values. In ‘Discussion’ section authors recognize some limitations in this sense. But the argument about the use of self-reports is clearly insufficient. I think they can try to justify the opportunity of the use of one-question measures, and in which sense, those measures are precise (reliable and valid) to assess a target variable. This is relevant because of any argument about LE with or without depression / anxiety can be understand as arbitrary.

Thank you for providing us with such a useful comment. The diagnosis of anxiety and depression was more than just a question. We provided more information about how the research participants diagnosed with depression and anxiety in the manuscript (line 161-172). We also mentioned that for the sampling method (reference 37) and some details of the surveys (line 125-137).

Point 4: Most relevant data are presented in percentages. Those data are discussed about increments/decrements between the two years when data were collected, comparing sex / genders, rural / urban… I miss some statistical coefficient (OR, chi squared…) to stablish when, comparing percentages, they are really different (statistical significance).

Thank you for your comments.

i:) In the manuscript, we compared the gender difference and urban-rural difference through the estimation of DAFLE, and we calculated the 95% confidence interval of DAFLE. we did linear regression analysis to test whether the DAFLE/LE decreased with age. the results show that all of P values were less than 0.05, which indicated that with older age and over time, DAFLE/LE gradually decreased (line 200-204). 

ii:) DAFLE/LE as a relative indicator reflects the percentage of healthy life years (without depression or anxiety) in total life expectancy. In order to explain the difference of man-women, urban-rural areas and 2013-2018 for DAFLE/LE, statistical method of the Wilcoxon Signed Ranks test was used. We added the description of this part to the manuscript (line 200-204).

Point 5: When data are discussed, one of the points is about women have more LE and they are more anxious / depressed. I could not found a specific data about specific prevalence rate of anxiety and depression. It is well-known women attain higher rates than men in both disorders and I suspect this data is coincident with their surveys. Manuscript statement about how women have higher LE and higher anxiety / depression levels and, in this sense, women will live more time with anxiety / depression sound tautological. I am not sure there is a direct correspondence (correlation) between LE and the presence of emotional negative feeling.

Thank you very much for your valuable comments. We agree with your review: as you said, it is well-known women attain higher rates than men in both disorders, which supports the findings that women have higher DAFLE/LE than men. We have added the prevalence data from the two surveys when describing the arguments for sex differences in the discussion.

Based on the comments made by the reviewer, we reviewed the manuscript carefully. We believe that it's different about the two descriptions: women have higher LE and DAFLE than men and women will spend more time in depression or anxiety. The results revealed by absolute numbers and relative indicator are different. The higher LE and DAFLE of women does not determine whether the DAFLE/LE of women is higher or lower than that of men. In other word, in our study, life expectancy without depression and anxiety for women is longer than that of men, but women are more likely to spend a higher percentage of their remaining lives with depression or anxiety than men. In fact, there were many similar studies that have reported healthy life expectancy and its proportion in life expectancy, such as healthy life expectancy without dementia (doi:10.1159/000381848) and healthy life expectancy without depression (doi:10.1590/S1518-8787.2016050005900).

Point 6: Finally, result implications are discussed about the need of suitable health services and resources. I agree with authors about this necessity. But I think data can also be discussed in how people deal with their lives and how emotional negative feelings are present across life span and the need for a new educational paradigm that includes psychological resources related with the use of efficient emotion regulation strategies. This is especially relevant if it is true that those negative feelings are predicting the initial onset of anxiety and depression (e.g., doi:10.3390/ijerph17196984)

Thanks. We agree with your comments and we have followed your suggestion. We have carefully read the article you recommended to us, we have quoted this article and added some necessary information to the discussion section of the manuscript (line 297-304), such as how do people with depression or anxiety deal with their lives. The discussion section in the manuscript had covered possible effective management strategies for depression and anxiety (line 308-315).

Sincerely,

Yuhang Wu

Round 2

Reviewer 2 Report

The authors have made the appropriate modifications. Manuscript can be published.

Reviewer 4 Report

I think authors have appropriately answered all questions and commentaries. Thanks for their efforts (and comprehensiveness).
I consider manuscript is now ready to be accepting.